# Form factor of local operators in the generalized algebraic Bethe ansatz

G. Kulkarni[1]

Univ Lyon, ENS de Lyon, Univ Claude Bernard Lyon 1, CNRS, Laboratoire de Physique,
F-69342 Lyon, France

N. A. Slavnov[2]

Steklov Mathematical Institute of Russian Academy of Sciences, Moscow, Russia

**Abstract**

We consider an $XYZ$ spin chain within the framework of the generalized algebraic
Bethe ansatz. We study form factors of local operators corresponding to the singlet states
in the limit of free fermions. We obtain explicit representations for these form factors.

**Key words:** Generalized algebraic Bethe ansatz, Bethe vectors, scalar products, form factors.

## 1   Introduction

In [1], we studied scalar products of Bethe vectors in the framework of the generalized algebraic Bethe ansatz [2]. This work is a continuation of the paper [1]. Here we calculate form factors of local spin operators in an $XYZ$ Heisenberg chain.

A Hamiltonian of the periodic $XYZ$ chain [3] has the following form:

$$H = \sum_{j=1}^{N} \Big( J_x \sigma_j^x \sigma_{j+1}^x + J_y \sigma_j^y \sigma_{j+1}^y + J_z \sigma_j^z \sigma_{j+1}^z \Big), \qquad \sigma_{N+1}^{x,y,z} = \sigma_1^{x,y,z}. \tag{1.1}$$

It acts in a Hilbert space $\mathcal{H} = \mathcal{H}_1 \otimes \mathcal{H}_2 \otimes \cdots \otimes \mathcal{H}_N$, where each $\mathcal{H}_k \cong \mathbb{C}^2$. The spin-1/2 operators $\sigma_k^{x,y,z}$ are Pauli matrices acting non-trivially in $\mathcal{H}_k$. Numerical coefficients $J_{x,y,z}$ play the role of interaction strength along the axis $x$, $y$, and $z$. We assume that the number of sites $N$ is even.

To study the $XYZ$ chain, the generalized algebraic Bethe ansatz is used [2]. This is because the $XYZ$ model is equivalent to an 8-vertex model [4–7] and has an 8-vertex $R$-matrix. However, when calculating the form factors of local operators, we use the same

---

[1]giridhar.kulkarni@ens-lyon.fr
[2]nslavnov@mi-ras.ru

scheme as in the models solved by the standard algebraic Bethe ansatz [8–11]. First, we express the local spin operators in terms of the monodromy matrix elements. This is achieved using the quantum inverse problem [12–14]. Second, we calculate the actions of the elements of the monodromy matrix on the Bethe vectors [15]. In the third stage, it is necessary to calculate the resulting scalar products [1, 16]. After performing the above steps, we obtain representations for the form factors of local operators in the form of some sums of scalar products. We investigate these representations in the present paper.

In the scheme described above, the last stage is the most difficult. In models with a 6-vertex $R$-matrix, the scalar products of Bethe vectors are well studied [10–12, 17]. However, the scalar products in models solvable by the generalized algebraic Bethe ansatz have been studied to a much lesser extent. To date, there is only one method that is based on reducing this problem to solving a system of linear equations [18]. Using this method, determinant representations were obtained for scalar products, in which both vectors depend on the same number of parameters [16]. Explicit representations for scalar products of a more general form have so far been obtained in [1] only for singlet states in the special case when the $XYZ$ chain is equivalent to free fermions ($XY$ model [19]). Therefore, in this paper, we calculate the form factors of local operators only in this particular case.

The $XY$ model is well studied. Many different methods allow one to calculate not only the form factors of local operators but also the correlation functions in this model [19–27]. Therefore, we do not set ourselves the goal of obtaining new results. Our main goal is to show the fundamental applicability of the generalized algebraic Bethe ansatz to the calculation of the form factors. In the future, we plan to generalize this approach to the general case of the $XYZ$ chain.

The paper is organized as follows. In section 2, we give a brief description of the generalized algebraic Bethe ansatz. In section 3, we define the form factors of local spin operators and describe their calculation scheme. In section 4, we present the main results of the paper. Finally, in section 5, we give examples of calculating the form factors. In particular, we show that singlet states have zero magnetization in section 5.1. In section 5.2, we compute transversal form factors.

At the end of this paper we have collected basic information about Jacobi theta functions in appendix A. In appendix B, we describe numeric coefficients that arise when the monodromy matrix elements act on Bethe vectors. Finally, in appendix C, we give explicit representations for the scalar products of Bethe vectors.

## 2 Generalized algebraic Bethe ansatz for the XYZ model

In this section, we provide basic information about the description of the $XYZ$ model by the generalized algebraic Bethe ansatz. The reader can get acquainted with this method in more detail in [2, 16].

### 2.1 $R$-matrix and monodromy matrix

The $R$-matrix of the 8-vertex model has the following form:

$$R(u) = \begin{pmatrix} \mathsf{a}(u) & 0 & 0 & \mathsf{d}(u) \\ 0 & \mathsf{b}(u) & \mathsf{c}(u) & 0 \\ 0 & \mathsf{c}(u) & \mathsf{b}(u) & 0 \\ \mathsf{d}(u) & 0 & 0 & \mathsf{a}(u) \end{pmatrix}, \tag{2.1}$$

where

$$a(u) = \frac{2\theta_4(\eta|2\tau)\,\theta_1(u+\eta|2\tau)\,\theta_4(u|2\tau)}{\theta_2(0|\tau)\,\theta_4(0|2\tau)},$$

$$b(u) = \frac{2\theta_4(\eta|2\tau)\,\theta_4(u+\eta|2\tau)\,\theta_1(u|2\tau)}{\theta_2(0|\tau)\,\theta_4(0|2\tau)},$$

$$c(u) = \frac{2\theta_1(\eta|2\tau)\,\theta_4(u+\eta|2\tau)\,\theta_4(u|2\tau)}{\theta_2(0|\tau)\,\theta_4(0|2\tau)},$$

$$d(u) = \frac{2\theta_1(\eta|2\tau)\,\theta_1(u+\eta|2\tau)\,\theta_1(u|2\tau)}{\theta_2(0|\tau)\,\theta_4(0|2\tau)}.$$

(2.2)

The definition of the Jacobi theta functions is given in appendix A. The parameters $\eta$ and $\tau$ are related to the coefficients $J_{x,y,z}$ (see (2.7)).

Within the framework of the generalized algebraic Bethe ansatz, Hamiltonian (1.1) is constructed from the monodromy matrix $\mathcal{T}(u)$. This is a $2 \times 2$ matrix, acting in an auxiliary space $\mathcal{H}_0 \cong \mathbb{C}^2$

$$\mathcal{T}(u) = \begin{pmatrix} A(u) & B(u) \\ C(u) & D(u) \end{pmatrix}.$$

(2.3)

The matrix elements are operators depending on the complex $u$ and acting on the Hilbert space $\mathcal{H}$. The monodromy matrix of the $XYZ$ chain of the length $N$ is equal to a product of the $R$-matrices acting in $\mathcal{H}_0 \otimes \mathcal{H}_k$:

$$\mathcal{T}(u) = R_{01}(u - \xi_1)R_{02}(u - \xi_2)\cdots R_{0N}(u - \xi_N),$$

(2.4)

where complex parameters $\xi_k$ are called inhomogeneities. To construct the Hamiltonian (1.1) we need only a homogeneous case $\xi_k = 0$, $k = 1, \ldots, N$. Then

$$H = \frac{2\theta_1(\eta|\tau)}{\theta_1'(0|\tau)}\frac{\mathrm{d}}{\mathrm{d}u}\log\mathsf{T}(u)\Big|_{u=0} - \frac{\theta_1'(\eta|\tau)}{\theta_1'(0|\tau)}N\mathbf{1},$$

(2.5)

where $\mathbf{1}$ is the identity operator, and

$$\mathsf{T}(u) = \mathrm{tr}\mathcal{T}(u) = A(u) + D(u).$$

(2.6)

The operator $\mathsf{T}(u)$ is called a transfer matrix. Eigenvectors of this operator coincide with the ones of the Hamiltonian and other integrals of motion.

The coupling constants of the Hamiltonian have the following representation:

$$J_x = \frac{\theta_4(\eta|\tau)}{\theta_4(0|\tau)}, \quad J_y = \frac{\theta_3(\eta|\tau)}{\theta_3(0|\tau)}, \quad J_z = \frac{\theta_2(\eta|\tau)}{\theta_2(0|\tau)}.$$

(2.7)

In the present paper, we focus on the case $\eta = 1/2$, which corresponds to $J_z = 0$. The corresponding model is called an $XY$ chain. It is equivalent to free fermions.

Although only a homogeneous case is needed to construct the Hamiltonian of the $XYZ$ chain, in what follows we will consider a more general inhomogeneous model (2.4) with arbitrary complex inhomogeneities $\xi_k$. We emphasize, however, that we do this solely for reasons of generality. In all the formulas below, the homogeneous limit is trivial.

## 2.2 Special notation

Before moving on, we introduce some new notation. From now on, we omit the modular parameter in the notation of theta functions whenever it is equal to $\tau$, namely, $\theta_a(\cdot) \equiv \theta_a(\cdot|\tau)$.

Let us also introduce two functions that will be often used below

$$f(u,v) = \frac{\theta_1(u-v+\eta)}{\theta_1(u-v)}, \qquad h(u,v) = \frac{\theta_1(u-v+\eta)}{\theta_1(\eta)}. \tag{2.8}$$

In what follows, we will constantly deal with sets of complex variables. We denote these sets by a bar: $\bar{u} = \{u_1, \ldots, u_m\}$, $\bar{v} = \{v_1, \ldots, v_n\}$, etc. We also introduce special subsets $\bar{u}_j = \bar{u} \setminus \{u_j\}$, $\bar{u}_{j,k} = \bar{u} \setminus \{u_j, u_k\}$ and so on.

To make the formulas more compact we use a shorthand notation for products of functions $f(u,v)$ and theta functions. Namely, if the function $f$ depends on a set (or two sets) of variables, this means that one should take the product over the corresponding set. For example,

$$f(u_j, \bar{u}_j) = \prod_{\substack{u_l \in \bar{u} \\ l \neq j}} f(u_j, u_l), \quad f(\bar{v}, \bar{u}) = \prod_{\substack{u_l \in \bar{u} \\ v_k \in \bar{v}}} f(v_k, u_l) \qquad \text{etc.} \tag{2.9}$$

Similarly,

$$\theta_2(u - \bar{v}) = \prod_{v_k \in \bar{v}} \theta_2(u - v_k), \qquad \theta_1(u_j - \bar{u}_j) = \prod_{\substack{u_l \in \bar{u} \\ l \neq j}} \theta_1(u_j - u_l), \qquad \text{etc.} \tag{2.10}$$

By definition, any product over the empty set is equal to 1. A double product is equal to 1 if at least one of the sets is empty.

### 2.2.1 Bethe vectors

In this paper, the explicit form of Bethe vectors is not essential. Therefore, we omit the details of their construction. The reader can find these details in [2, 16]. We only note that the Bethe vectors are constructed using a special gauge transformation of the monodromy matrix.

We denote Bethe vectors by $|\hat{\Psi}_n^\nu(\bar{u})\rangle$. They belong to the Hilbert space $\mathcal{H}$: $|\hat{\Psi}_n^\nu(\bar{u})\rangle \in \mathcal{H}$. Bethe vectors are parameterized be a set of complex numbers $\bar{u} = \{u_1, \ldots, u_n\}$ and an integer $\nu$. In the case of free fermions, $\nu \in \mathbb{Z}_4$. If these parameters are related by a system of Bethe equations and a sum rule (see below), then this vector is an eigenvector of the transfer matrix. We call it an on-shell Bethe vector in this case. Otherwise, the Bethe vector is called off-shell.

Let us introduce

$$\chi_\nu(z) = (-1)^n e^{i\pi\nu/2} a(z) + e^{-i\pi\nu/2} d(z), \tag{2.11}$$

where

$$a(z) = \theta_2(z - \bar{\xi}), \qquad d(z) = \theta_1(z - \bar{\xi}). \tag{2.12}$$

Then Bethe equations have the following form:

$$\chi_\nu(u_j) = 0, \qquad j = 1, \ldots, n. \tag{2.13}$$

Assume also that the parameters $\bar{u}$ and $\nu$ satisfy the sum rule:

$$2 \sum_{j=1}^n u_j = \sum_{k=1}^N \xi_k + \frac{n}{2} + \nu\tau + \nu_1, \tag{2.14}$$

where $\nu_1$ takes integer values.

If the conditions (2.13) and (2.14) are fulfilled, then

$$\mathsf{T}(z)|\hat{\Psi}_n^\nu(\bar{u})\rangle = T_\nu(z|\bar{u})|\hat{\Psi}_n^\nu(\bar{u})\rangle, \tag{2.15}$$

where the transfer matrix eigenvalue $T_\nu(z|\bar{u})$ is

$$T_\nu(z|\bar{u}) = \chi_\nu(z)f(z,\bar{u}). \tag{2.16}$$

In the $XYZ$ model with a rational value of $\eta$, there is a degeneracy of the spectrum [28–30]. In particular, for $\eta = 1/2$, the degeneracy is due to the presence of roots of Bethe equations differing from each other by $1/2$. Let us define the following mapping over the fundamental domain $z \in \mathbb{C}/(\mathbb{Z} + \tau\mathbb{Z})$

$$z^* = z + \frac{(-1)^\epsilon}{2}, \tag{2.17}$$

where $\epsilon = 0$ if $0 \le \Re z < \frac{1}{2}$ and $\epsilon = 1$ otherwise. It is easy to check that $\chi_\nu(z^*) = (-1)^\nu \chi_\nu(z)$ (recall that we consider the chain of even length $N$). Therefore, if $u_j$ is a root $\chi_\nu(z)$, then $u_j^*$ is also a root $\chi_\nu(z)$. We will call $u_j$ and $u_j^*$ twins.

In what follows, we will work only with twin-free on-shell Bethe vectors that correspond to singlet eigenstates. Consideration of vectors with twins requires a special study (see e.g. [31–34]).

For singlet states, $n = N/2$. In addition, we must require that there are no twins in the set $\bar{u}$, that is, $u_j \ne u_k \pm 1/2$ for any $u_j, u_k \in \bar{u}$.

**Proposition 2.1.** *Let $\bar{u}$ be a twin-free set of the roots of Bethe equations (2.13) satisfying the sum rule (2.14). Then*

$$\chi_\nu(z) = (-1)^n e^{-2\pi i\nu(z-\xi_p)+\pi i\nu/2} a(\xi_p) \frac{\theta_1(z-\bar{u})\theta_2(z-\bar{u})}{\theta_1(\xi_p-\bar{u})\theta_2(\xi_p-\bar{u})}, \tag{2.18}$$

*where $\xi_p$ is any of inhomogeneities.*

*Proof.* Since equation (2.11) is an elliptic polynomial of degree $N$, it has $N$ roots in the fundamental domain. First, these are the roots $\bar{u} = \{u_1, \ldots, u_n\}$. Second, these are their twins $\bar{u}^* = \{u_1^*, \ldots, u_n^*\}$. Then

$$\theta_1(z - \bar{u}^*) = (-1)^{n'}\theta_2(z - \bar{u}), \tag{2.19}$$

where $n'$ is the number of roots $u_j$ with $0 \le \Re u_j < 1/2$.

Consider a function

$$\mathsf{f}(z) = \frac{\chi_\nu(z)}{\theta_1(z-\bar{u})\theta_1(z-\bar{u}^*)} = \frac{(-1)^{n'}\chi_\nu(z)}{\theta_1(z-\bar{u})\theta_2(z-\bar{u})}. \tag{2.20}$$

Obviously, $\mathsf{f}(z)$ is an entire function of $z$. Due to (A.3) we have $\mathsf{f}(z+1) = \mathsf{f}(z)$, and

$$\mathsf{f}(z + \tau) = e^{-2\pi i\nu\tau}\mathsf{f}(z), \tag{2.21}$$

where we used the sum rule. Hence,

$$(-1)^{n'}\frac{e^{2\pi i\nu z}\chi_\nu(z)}{\theta_1(z-\bar{u})\theta_2(z-\bar{u})} = C, \tag{2.22}$$

where $C$ is a constant. To find this constant, we set $z = \xi_p$, where $\xi_p$ is an arbitrary inhomogeneity. Then $d(\xi_p) = 0$, and $\chi_\nu(\xi_p) = (-1)^n e^{i\pi\nu/2} a(\xi_p)$. Thus,

$$C = (-1)^{n'+n}\frac{e^{2\pi i\nu\xi_p+i\pi\nu/2}a(\xi_p)}{\theta_1(\xi_p-\bar{u})\theta_2(\xi_p-\bar{u})}, \tag{2.23}$$

and we arrive at (2.18). $\qquad\square$

The Bethe vectors also depend on two parameters of the gauge transformation $s$ and $t$, which are arbitrary complex numbers. We do not explicitly indicate this dependence in the notation. In what follows, we will use the following combinations:

$$x = \frac{s+t}{2}, \qquad y = \frac{s-t}{2}. \tag{2.24}$$

We also set

$$s_k = s + \frac{k}{2}, \qquad t_k = t + \frac{k}{2}, \qquad x_k = \frac{s_k + t_k}{2}. \tag{2.25}$$

For the singlet eigenstate, the dependence on $s$ and $t$ is present only in the scalar factor

$$|\hat{\Psi}_n^\nu(\bar{u})\rangle = \varphi(s,t)|\widetilde{\Psi}_n^\nu(\bar{u})\rangle, \tag{2.26}$$

where $|\widetilde{\Psi}_n^\nu(\bar{u})\rangle$ does not depend on the gauge parameters. Therefore, the dependence on $s$ and $t$ disappears in normalized expressions.

To calculate the form factors, we also need dual Bethe vectors $\langle\hat{\Psi}_n^\nu(\bar{u})|$. They belong to the dual space $\mathcal{H}^*$ and are arranged in a completely similar way to the Bethe vectors described above. In particular, they are left eigenvectors of the transfer matrix

$$\langle\hat{\Psi}_n^\nu(\bar{u})|\mathsf{T}(z) = T_\nu(z|\bar{u})\langle\hat{\Psi}_n^\nu(\bar{u})|, \tag{2.27}$$

if the parameters $\bar{u}$ and $\nu$ satisfy conditions (2.13) and (2.14). Then we call them dual on-shell Bethe vectors.

## 3  Form factors

In this section, we move on to the form factors of local operators, which is the main topic of this paper. We call a form factor a matrix element of the form

$$\mathcal{F}_{a,p}^{\nu,\lambda}(\bar{v}|\bar{u}) = \mathcal{N}_n^\nu(\bar{v})\,\langle\hat{\Psi}_n^\nu(\bar{v})|\sigma_p^a|\hat{\Psi}_n^\lambda(\bar{u})\rangle, \qquad a \in \{x,y,z\}. \tag{3.1}$$

Here $\langle\hat{\Psi}_n^\nu(\bar{v})|$ and $|\hat{\Psi}_n^\lambda(\bar{u})\rangle$ are, respectively, the left and right eigenstates of the transfer matrix. Recall that in this paper, we consider only the singlet part of the spectrum, that is, twin-free states. The normalization factor is chosen in a non-standard way

$$\mathcal{N}_n^\nu(\bar{v}) = \left(\langle\hat{\Psi}_n^\nu(\bar{v})|\hat{\Psi}_n^\nu(\bar{v})\rangle\right)^{-1}. \tag{3.2}$$

This is because it was in this normalization that the scalar products were calculated in [1, 16]

$$\mathbf{S}_{n,m}^{\nu,\mu}(\bar{v}|\bar{w}) = \mathcal{N}_n^\nu(\bar{v})\langle\hat{\Psi}_n^\nu(\bar{v})|\hat{\Psi}_m^\mu(\bar{w})\rangle. \tag{3.3}$$

Here $\langle\hat{\Psi}_n^\nu(\bar{v})|$ is an on-shell Bethe vector, while $|\hat{\Psi}_m^\mu(\bar{w})\rangle$ is an arbitrary off-shell vector.

Since the Bethe vectors in (3.1) are not normalized, the form factor may depend on the parameters of the gauge transformation $s$ and $t$. However, when calculating correlation functions, we usually deal with expressions that are quadratic in form factors, for instance,

$$\mathcal{F}_{a,p}^{\nu,\lambda}(\bar{v}|\bar{u})\mathcal{F}_{a,p'}^{\lambda,\nu}(\bar{u}|\bar{v}) = \mathcal{N}_n^\nu(\bar{v})\mathcal{N}_n^\lambda(\bar{u})\,\langle\hat{\Psi}_n^\nu(\bar{v})|\sigma_p^a|\hat{\Psi}_n^\lambda(\bar{u})\rangle\,\langle\hat{\Psi}_n^\lambda(\bar{u})|\sigma_{p'}^a|\hat{\Psi}_n^\nu(\bar{v})\rangle. \tag{3.4}$$

We see that such expressions turn out to be normalized in the standard way. Therefore, they should not depend on the parameters of the gauge transformation. Such a disappearance of the dependence on $s$ and $t$ is one of the criteria for the correctness of the results obtained.

Using the solution of the inverse scattering problem[3] [12–14]

$$\sigma_p^a = \left( \prod_{i=1}^{p} \mathsf{T}^{-1}(\xi_i) \right) \operatorname{tr} \left( \sigma^a \mathcal{T}(\xi_p) \right) \left( \prod_{i=1}^{p-1} \mathsf{T}(\xi_i) \right), \tag{3.5}$$

we can reduce the form factors of local operators to the form factors of the monodromy matrix entries

$$\mathcal{F}_{a,p}^{\nu,\lambda}(\bar{v}|\bar{u}) = \frac{\prod_{i=1}^{p-1} T_\lambda(\xi_i|\bar{u})}{\prod_{i=1}^{p} T_\nu(\xi_i|\bar{v})} \mathbf{F}_a^{\nu,\lambda}(\bar{v}|\bar{u}). \tag{3.6}$$

Here $\mathbf{F}_a^{\nu,\lambda}(\bar{v}|\bar{u})$ is represented as

$$\mathbf{F}_a^{\nu,\lambda}(\bar{v}|\bar{u}) = \mathcal{N}_n^\nu(\bar{v}) \langle \hat{\Psi}_n^\nu(\bar{v})| \operatorname{tr} \left( \sigma^a \mathcal{T}(\xi_p) \right) |\hat{\Psi}_n^\lambda(\bar{u})\rangle. \tag{3.7}$$

We do not indicate dependence on $\xi_p$ in the notation $\mathbf{F}_a^{\nu,\lambda}(\bar{v}|\bar{u})$ for brevity.

In its turn, the action of the operators $\operatorname{tr} \left( \sigma^a \mathcal{T}(\xi_p) \right)$ on Bethe vectors $|\hat{\Psi}_n^\lambda(\bar{u})\rangle$ was calculated in [15]. For free fermions, it has the following form:

$$\operatorname{tr} \left( \sigma^a \mathcal{T}(w_{n+1}) \right) |\hat{\Psi}_n^\lambda(\bar{w}_{n+1})\rangle = \sum_{\mu=0}^{3} \left\{ \sum_{k=1}^{n+1} \mathbf{W}_{a;0}^{(\lambda,\mu)}(w_{n+1}, w_k)|\hat{\Psi}_n^\mu(\bar{w}_k)\rangle \right.$$

$$\left. + \sum_{k>l}^{n+1} \mathbf{W}_{a;-}^{(\lambda-\mu)}(w_{n+1}, w_k, w_l)|\hat{\Psi}_{n-1}^\mu(\bar{w}_{k,l})\rangle + \mathbf{W}_{a;+}^{(\lambda-\mu)}(w_{n+1})|\hat{\Psi}_{n+1}^\mu(\bar{w})\rangle \right\}. \tag{3.8}$$

Here $w_{n+1} = \xi_p$ and $\bar{w}_{n+1} = \bar{u}$. The numerical coefficients $\mathbf{W}_{a;0}^{(\lambda,\mu)}$ and $\mathbf{W}_{a;\pm}^{(\lambda-\mu)}$ were calculated in [15]. We give their explicit form in appendix B.

Equation (3.8) immediately allows us to reduce the form factors $\mathbf{F}_a^{\nu,\lambda}(\bar{v}|\bar{u})$ to linear combinations of the scalar products

$$\mathbf{F}_a^{\nu,\lambda}(\bar{v}|\bar{u}) = \sum_{\mu=0}^{3} \left\{ \sum_{k=1}^{n+1} \mathbf{W}_{a;0}^{(\lambda,\mu)}(w_{n+1}, w_k) \mathbf{S}_{n,n}^{\nu,\mu}(\bar{v}|\bar{w}_k) \right.$$

$$\left. + \sum_{k>l}^{n+1} \mathbf{W}_{a;-}^{(\lambda-\mu)}(w_{n+1}, w_k, w_l) \mathbf{S}_{n,n-1}^{\nu,\mu}(\bar{v}|\bar{w}_{k,l}) + \mathbf{W}_{a;+}^{(\lambda-\mu)}(w_{n+1}) \mathbf{S}_{n,n+1}^{\nu,\mu}(\bar{v}|\bar{w}) \right\}. \tag{3.9}$$

The scalar products were calculated in [1, 16]. Their explicit form is given in appendix C. Thus, the calculation of form factors is reduced to the substitution of known expressions into formula (3.9).

## 4 Main results

In this section, we present the main results of the paper. Since the $XY$ model is planar, the longitudinal and transverse form factors are fundamentally different quantities. Therefore, we are considering these two types of form factors separately.

---

[3]In [12–14], the monodromy matrix was defined using the opposite ordering of the $R$-matrices. Therefore, formula (3.5) is somewhat different from the solution of the quantum inverse problem given in the above-mentioned papers.

## 4.1 Longitudinal form factors

Non-zero form factors $\mathcal{F}_{z,p}^{\nu,\lambda}(\bar{v}|\bar{u})$ are those for which $\lambda = \nu$, $\nu = 0, 1$, and $\bar{u} = \{\bar{v}_k, v_k^*\}$, $k = 1, \ldots, n$. In other words, the set $\bar{u}$ contains the twin of some element $v_k$, while all other elements are the same as elements of $\bar{v}$. We can set $u_n = v_n^*$ and $\bar{u}_n = \bar{v}_n$ without loss of generality. Then

$$
\mathcal{F}_{z,p}^{\nu,\nu}(\bar{v}|\{\bar{v}_n, v_n^*\}) = \left( \prod_{j=1}^{p-1} \frac{1}{f(\xi_j, v_n) f(v_n, \xi_j)} \right)
$$
$$
\times \frac{2(-1)^n e^{-i\pi\nu/2} \theta_1'(0)\theta_2(0)}{\mathcal{V}_n f(v_n, \bar{v}_n) f(\bar{v}_n, v_n) \theta_4(0|2\tau)} \frac{\theta_2(v_n + y)}{\theta_1(v_n + y)} \frac{\theta_4(2v_n - 2\xi_p|2\tau)}{\theta_2^2(v_n - \xi_p)}, \quad (4.1)
$$

where

$$
\mathcal{V}_k = \frac{d}{dz} \log \frac{a(z)}{d(z)} \Big|_{z=v_k}. \quad (4.2)
$$

We see that the individual form factor depends on the gauge transformation parameter $y$. However, this dependence disappears if we consider the quadratic combination of form factors:

$$
\mathcal{F}_{z,p}^{\nu,\nu}(\bar{v}|\{\bar{v}_n, v_n^*\}) \mathcal{F}_{z,p'}^{\nu,\nu}(\{\bar{v}_n, v_n^*\}|\bar{v}) = \left( \prod_{j=p-1}^{p'-1} f(\xi_j, v_n) f(v_n, \xi_j) \right)
$$
$$
\times 4 e^{-i\pi\nu} \left( \frac{\theta_1'(0)\theta_2(0)}{\mathcal{V}_n \theta_4(0|2\tau)} \right)^2 \frac{\theta_4(2v_n - 2\xi_p|2\tau)\theta_4(2v_n - 2\xi_{p'}|2\tau)}{\theta_2^2(v_n - \xi_p)\theta_1^2(v_n - \xi_{p'})}, \quad (4.3)
$$

where we set $p' \geq p$ for definiteness.

## 4.2 Transversal form factors

Non-zero form factors $\mathcal{F}_{x,p}^{\nu,\lambda}(\bar{v}|\bar{u})$ and $\mathcal{F}_{y,p}^{\nu,\lambda}(\bar{v}|\bar{u})$ occur for either $\nu = 0$, $\lambda = 1$ or $\nu = 1$, $\lambda = 0$. Then

$$
\mathcal{F}_{x,p}^{\nu,\lambda}(\bar{v}|\bar{u}) = -i\mu_0 \frac{i^{\mu_1} + (-i)^{\mu_1}}{2} \theta_4(0) \mathcal{F}_p^{\nu,\lambda}(\bar{v}|\bar{u}),
$$
$$
\mathcal{F}_{y,p}^{\nu,\lambda}(\bar{v}|\bar{u}) = \frac{i^{\mu_1} - (-i)^{\mu_1}}{2} \theta_3(0) \mathcal{F}_p^{\nu,\lambda}(\bar{v}|\bar{u}) \quad (4.4)
$$

where

$$
\mathcal{F}_p^{\nu,\lambda}(\bar{v}|\bar{u}) = \left( \prod_{j=1}^{p-1} \frac{f(\bar{u}, \xi_j)}{f(\bar{v}, \xi_j)} \right) e^{i\pi(\mu_0(s+\xi_p) - \tau/4 + \nu/2)} S^\nu(\bar{v}|\bar{u}) \frac{\theta_2(0)}{\theta_4^2(0|2\tau)} \frac{\theta_1(y + \bar{u})\theta_2(\xi_p - \bar{v})}{\theta_1(y + \bar{v})\theta_2(\xi_p - \bar{u})}. \quad (4.5)
$$

In these formulas, $\mu_0 = \lambda - \nu$, the function $S^\nu(\bar{v}|\bar{u})$ is given by (C.3), and

$$
\mu_1 = 2 \sum_{j=1}^n (v_j - u_j) + \mu_0 \tau. \quad (4.6)
$$

Due to the sum rule, $\mu_1$ is an integer. Thus, the form factor $\mathcal{F}_{x,p}^{\nu,\lambda}$ vanishes for $\mu_1$ odd, while $\mathcal{F}_{y,p}^{\nu,\lambda}$ vanishes for $\mu_1$ even.

We see that, as in the case above, we have a dependence on the gauge parameters. However, they are no longer included in the quadratic expressions. Indeed, setting for definiteness $p' \geq p$ we obtain

$$
\begin{aligned}
\mathcal{F}^{\nu,\lambda}_{x,p}(\bar{v}|\bar{u})\mathcal{F}^{\lambda,\nu}_{x,p'}(\bar{u}|\bar{v}) &= \frac{1+(-1)^{\mu_1}}{2}\theta_4^2(0)\mathfrak{F}^{\nu,\lambda}_{p,p'}(\bar{v}|\bar{u}), \\
\mathcal{F}^{\nu,\lambda}_{y,p}(\bar{v}|\bar{u})\mathcal{F}^{\lambda,\nu}_{y,p'}(\bar{u}|\bar{v}) &= \frac{1-(-1)^{\mu_1}}{2}\theta_3^2(0)\mathfrak{F}^{\nu,\lambda}_{p,p'}(\bar{v}|\bar{u}),
\end{aligned}
\tag{4.7}
$$

where

$$
\mathfrak{F}^{\nu,\lambda}_{p,p'}(\bar{v}|\bar{u}) = -i\left(\prod_{j=p-1}^{p'-1}\frac{f(\bar{v},\xi_j)}{f(\bar{u},\xi_j)}\right)e^{i\pi\mu_0(\xi_p-\xi_{p'})-i\pi\tau/2}\left(\frac{\theta_2(0)}{\theta_4^2(0|2\tau)}\right)^2
$$

$$
\times\, S^\nu(\bar{v}|\bar{u})S^\lambda(\bar{u}|\bar{v})\frac{\theta_2(\xi_p-\bar{v})\theta_2(\xi_{p'}-\bar{u})}{\theta_2(\xi_p-\bar{u})\theta_2(\xi_{p'}-\bar{v})}, \tag{4.8}
$$

and we used $\nu+\lambda=1$.

# 5 Examples of calculating form factors

In this section, we give two examples of calculating form factors. In the first example, we prove that the magnetization of any singlet state is zero. In the second example, we consider the transversal form factor.

The calculation of form factors is straightforward but rather tedious. Therefore, we do not provide all the details. Instead, we focus only on the part of the computation that requires some non-trivial steps.

## 5.1 Zero magnetization

We begin our consideration with a magnetization

$$
\mathcal{F}^{\nu,\nu}_{z,p}(\bar{v}|\bar{v}) = \mathcal{N}^\nu_n(\bar{v})\langle\hat{\Psi}^\nu_n(\bar{v})|\sigma^z_p|\hat{\Psi}^\nu_n(\bar{v})\rangle, \qquad p=1,\dots,N. \tag{5.1}
$$

Due to (3.6) we have

$$
\mathcal{F}^{\nu,\nu}_{z,p}(\bar{v}|\bar{v}) = \frac{1}{T_\nu(\xi_p|\bar{v})}\mathbf{F}^{\nu,\nu}_z(\bar{v}|\bar{v}), \tag{5.2}
$$

where

$$
\mathbf{F}^{\nu,\nu}_z(\bar{v}|\bar{v}) = \mathcal{N}^\nu_n(\bar{v})\langle\hat{\Psi}^\nu_n(\bar{v})|\big(A(\xi_p)-D(\xi_p)\big)|\hat{\Psi}^\nu_n(\bar{v})\rangle. \tag{5.3}
$$

Recall that $\xi_p$ is the inhomogeneity corresponding to the $p$th site.

In the process of calculations, the form factor $\mathbf{F}^{\nu,\nu+2}_z(\bar{v}|\bar{v})$ also naturally arises. Therefore, from the very beginning, we will consider two form factors

$$
\mathbf{F}^{\nu,\lambda}_z(\bar{v}|\bar{v}) = \mathcal{N}^\nu_n(\bar{v})\langle\hat{\Psi}^\nu_n(\bar{v})|\big(A(\xi_p)-D(\xi_p)\big)|\hat{\Psi}^\lambda_n(\bar{v})\rangle, \qquad \lambda=\nu,\ \nu+2. \tag{5.4}
$$

Note that, from a formal point of view, the Bethe equations (2.13) do not change when $\nu$ is replaced by $\nu+2$.

It is convenient to divide the form factor (5.4) into three parts, which correspond to the three terms on the rhs of (3.9):

$$
\mathbf{F}^{\nu,\lambda}_z(\bar{v}|\bar{v}) = \mathbf{F}^{\nu,\lambda}_{z,0}(\bar{v}|\bar{v}) + \mathbf{F}^{\nu,\lambda}_{z,+}(\bar{v}|\bar{v}) + \mathbf{F}^{\nu,\lambda}_{z,-}(\bar{v}|\bar{v}). \tag{5.5}
$$

We first consider $\mathbf{F}^{\nu,\lambda}_{z,0}(\bar{v}|\bar{v})$. Using (B.3) we obtain

$$\mathbf{F}^{\nu,\lambda}_{z,0}(\bar{v}|\bar{v}) = \sum_{\mu=0}^{3} \sum_{k=1}^{n+1} \frac{\hat{\alpha}^z_{\lambda-\mu}(\xi_p - w_k) f(w_k, \bar{w}_k)\left((-1)^n e^{i\pi\mu/2} a(w_k) - e^{-i\pi\mu/2} d(w_k)\right)}{4\theta_1(y+\xi_p) h(w_k, \xi_p)\theta_1(x)\theta_2(x)} \mathbf{S}^{\nu,\mu}_{n,n}(\bar{v}|\bar{w}_k).$$

(5.6)

Here $\bar{w} = \{\bar{v}, \xi_p\}$ so that $w_{n+1} = \xi_p$ and $\bar{w}_{n+1} = \bar{v}$.

Due to selection rule (C.1) $\mathbf{S}^{\nu,\nu+1}_{n,n} = \mathbf{S}^{\nu,\nu+3}_{n,n} = 0$. Besides, it is easy to show that $\hat{\alpha}^z_1(z) = \hat{\alpha}^z_3(z) = 0$, because $\alpha^z_l(z) = \alpha^z_{l+2}(z)$. Thus, we conclude that non-vanishing contributions to $\mathbf{F}^{\nu,\lambda}_{z,0}(\bar{v}|\bar{v})$ occur only for $\lambda = \nu$ and $\lambda = \nu + 2$. Therefore, we introduce the following linear combinations:

$$\mathbf{F}^{\nu;\epsilon}_{z,0}(\bar{v}|\bar{v}) = \mathbf{F}^{\nu,\nu}_{z,0}(\bar{v}|\bar{v}) + (-1)^\epsilon \mathbf{F}^{\nu,\nu+2}_{z,0}(\bar{v}|\bar{v}), \qquad \epsilon = 0, 1. \tag{5.7}$$

Then it is easy to see that

$$\mathbf{F}^{\nu;\epsilon}_{z,0}(\bar{v}|\bar{v}) = \sum_{k=1}^{n+1} \frac{\alpha^z_\epsilon(\xi_p - w_k) f(w_k, \bar{w}_k)\left((-1)^n e^{i\pi\nu/2} a(w_k) - e^{-i\pi\nu/2} d(w_k)\right)}{\theta_1(y+\xi_p) h(w_k, \xi_p)\theta_1(x)\theta_2(x)} \mathbf{S}^{\nu;1-\epsilon}_{n,n}(\bar{v}|\bar{w}_k),$$

(5.8)

where we used

$$\hat{\alpha}^z_0(z) + (-1)^\epsilon \hat{\alpha}^z_2(z) = 4\alpha^z_\epsilon(z). \tag{5.9}$$

One should distinguish two cases: $k = n+1$ and $k < n+1$. In the first case, $w_k = \xi_p$ and $\bar{w}_k = \bar{v}$. Then $d(w_k) = 0$ and $\mathbf{S}^{\nu;1-\epsilon}_{n,n}(\bar{v}|\bar{v}) = 1$ (see (C.3)–(C.4)). In the second case, $w_k = v_k$ and $\bar{w}_k = \{\bar{v}_k, \xi_p\}$. Then

$$(-1)^n e^{i\pi\nu/2} a(v_k) - e^{-i\pi\nu/2} d(v_k) = 2(-1)^n e^{i\pi\nu/2} a(v_k), \tag{5.10}$$

due to Bethe equations (2.13), and

$$\mathbf{S}^{\nu;1-\epsilon}_{n,n}(\bar{v}|\{\bar{v}_k, \xi_p\}) = \frac{\theta'_1(0) a(\xi_p) f(\xi_p, \bar{v})\theta_2(v_k - \xi_p + x_\epsilon)}{a(v_k)\mathcal{V}_k f(v_k, \bar{v}_k)\theta_2(x_\epsilon)\theta_2(v_k - \xi_p)}, \tag{5.11}$$

where $\mathcal{V}_k$ is given by (4.2).

Substituting these formulas into (5.8) we obtain

$$\mathbf{F}^{\nu;\epsilon}_{z,0}(\bar{v}|\bar{v}) = C_z \sum_{k=0}^{n} \mathcal{G}^z_{k;0}. \tag{5.12}$$

Here

$$C_z = \frac{e^{i\pi\nu/2} a(\xi_p) f(\bar{v}, \xi_p)\theta'_1(0)\theta_2^2(0)}{\theta_1(y+\xi_p)\theta_1(x_\epsilon)\theta_2^2(x_\epsilon)}, \tag{5.13}$$

$$\mathcal{G}^z_{0;0} = \theta_2(y+\xi_p)\frac{\theta_1^2(x_\epsilon)\theta_2(x_\epsilon)}{\theta'_1(0)\theta_2^2(0)}, \tag{5.14}$$

and

$$\mathcal{G}^z_{k;0} = -2\frac{\theta_2(y+\xi_p)\theta_1(x_\epsilon)\theta_1(v_k - \xi_p - x_\epsilon)\theta_2(v_k - \xi_p + x_\epsilon)}{\mathcal{V}_k \theta_1(v_k - \xi_p)\theta_2(v_k - \xi_p)\theta_2(0)}, \qquad k > 0. \tag{5.15}$$

We do the same with the other two contributions to the form factor. Namely, we introduce

$$\mathbf{F}^{\nu;\epsilon}_{\pm,0}(\bar{v}|\bar{v}) = \mathbf{F}^{\nu,\nu}_{\pm,0}(\bar{v}|\bar{v}) + (-1)^\epsilon \mathbf{F}^{\nu,\nu+2}_{\pm,0}(\bar{v}|\bar{v}), \qquad \epsilon = 0, 1. \tag{5.16}$$

Then using (B.6) and (B.10) we obtain

$$\mathbf{F}_{-,0}^{\nu;\epsilon}(\bar{v}|\bar{v}) = 2i^\epsilon \sum_{k>l}^{n+1} \frac{\beta_\epsilon^z(\xi_p, w_k, w_l)\omega_{kl}(\xi_p)}{\theta_1(y+\xi_p)\theta_2(0)\theta_1^2(x)\theta_2^2(x)}\mathbf{S}_{n,n-1}^{\nu,\epsilon}(\bar{v}|\bar{w}_{k,l}) \tag{5.17}$$

and

$$\mathbf{F}_{+,0}^{\nu;\epsilon}(\bar{v}|\bar{v}) = -i^\epsilon \frac{\theta_2(0)\gamma_\epsilon^z(\xi_p)}{2\theta_1(y+\xi_p)}\mathbf{S}_{n,n+1}^{\nu,\epsilon}(\bar{v}|\bar{w}), \tag{5.18}$$

where $\beta_\epsilon^z$, $\gamma_\epsilon^z$, and $\omega_{kl}(z)$ respectively are given by (B.4), (B.8), and (B.7). Recall also that $\bar{w} = \{\bar{v}, \xi_p\}$. To derive equations (5.17) and (5.18) we used $\beta_{l+2} = -\beta_l$ and $\gamma_{l+2} = -\gamma_l$, which results in

$$\hat{\beta}_1 + (-1)^\epsilon \hat{\beta}_3 = 4(-i)^\epsilon \beta_\epsilon, \qquad \hat{\gamma}_1 + (-1)^\epsilon \hat{\gamma}_3 = 4(-i)^\epsilon \gamma_\epsilon. \tag{5.19}$$

Using explicit representations for $\beta_\epsilon^z$ and the scalar product $\mathbf{S}_{n,n-1}^{\nu,\epsilon}$ we obtain

$$\mathbf{F}_{z,-}^{\nu;\epsilon}(\bar{v}|\bar{v}) = C_z \sum_{k=1}^n \mathcal{G}_{k;-}^z, \tag{5.20}$$

where

$$\mathcal{G}_{k;-}^z = \frac{\theta_1(t_\epsilon - \xi_p)\theta_1(v_k + s_\epsilon)\theta_1(v_k - \xi_p - x_\epsilon)}{\mathcal{V}_k\theta_2(v_k - \xi_p)\theta_1(v_k + y)}. \tag{5.21}$$

Similarly, using representations for $\gamma_\epsilon^z$ and the scalar product $\mathbf{S}_{n,n+1}^{\nu,\epsilon}$ we find

$$\mathbf{F}_{z,+}^{\nu;\epsilon}(\bar{v}|\bar{v}) = C_z \sum_{k=0}^n \mathcal{G}_{k;+}^z. \tag{5.22}$$

Here

$$\mathcal{G}_{0;+}^z = e^{-i\pi\nu/2} \frac{\theta_2(x_\epsilon)\theta_1(s_\epsilon + \xi_p)\theta_1(t_\epsilon - \xi_p)}{\theta_1'(0)\theta_2(y+\xi_p)} \frac{d(-y^*)}{\chi_\nu(-y^*)}, \tag{5.23}$$

with $y^* = y - 1/2$, and

$$\mathcal{G}_{k;+}^z = \frac{\theta_1(s_\epsilon + \xi_p)\theta_1(v_k - t_\epsilon)\theta_2(v_k - \xi_p + x_\epsilon)}{\mathcal{V}_k\theta_1(v_k - \xi_p)\theta_2(v_k + y)}, \qquad k > 0. \tag{5.24}$$

Deriving contributions (5.20) and (5.22) we used the fact that $a(v_k)d(v_l) - a(v_l)d(v_k) = 0$ due to the Bethe equations. Thanks to this property, all double sums that are initially present in the corresponding expressions turn into single ones.

Combining all the contributions above and using identities (A.10), (A.11) we arrive at the following representation for the form factor:

$$\mathbf{F}_z^{\nu;\epsilon}(\bar{v}|\bar{v}) = C_z \left\{ \mathcal{G}_{0;0}^z + \mathcal{G}_{0;+}^z + \sum_{k=1}^n \mathcal{G}_k^z \right\}, \tag{5.25}$$

where

$$\mathcal{G}_k^z = \frac{\theta_2(x_\epsilon)\theta_1(y+\xi_p)}{\theta_2(0)\mathcal{V}_k}\Big(\Omega(v_k - 1/2) - \Omega(v_k)\Big), \tag{5.26}$$

and

$$\Omega(z) = \frac{\theta_2(z - \xi_p - x_\epsilon)\theta_2(z - \xi_p + x_\epsilon)\theta_1(z + y)}{\theta_1(z - \xi_p)\theta_2(z - \xi_p)\theta_2(z + y)}. \tag{5.27}$$

*Remark.* Obviously, $\Omega(z+1) = \Omega(z)$. Therefore, we can replace $\Omega(v_k - 1/2)$ with $\Omega(v_k + 1/2)$ in (5.26). We have chosen the minus sign for definiteness only.

Let us prove that the combination in braces in (5.25) vanishes. For this, we consider a contour integral

$$J = e^{-i\pi\nu/2} \frac{\theta_2(x_\epsilon)\theta_1(y + \xi_p)}{2\pi i \theta_2(0)} \oint \Omega(z) \frac{d(z)}{\chi_\nu(z)} \, dz, \tag{5.28}$$

where the integration is taken along the boundary of the fundamental domain. Due to the periodicity of the integrand, we conclude that $J = 0$. On the other hand, the integral is equal to the sum of the residues in the poles within the integration contour. First of all, we have poles in the roots of Bethe equations $z = v_k$ and their twins $z = v_k^*$, $k = 1, \ldots, n$. It is easy to see that these poles give us the sum of $\mathcal{G}_k^z$. There are two additional poles in $z = -y + 1/2 = -y^*$ and $z = \xi_p + 1/2$ (the pole at $z = \xi_p$ is compensated by the zero of the function $d(z)$, since $d(\xi_p) = 0$). Thus, we obtain

$$\sum_{k=1}^{n} \mathcal{G}_k^z = -e^{-i\pi\nu/2} \frac{\theta_2(x_\epsilon)\theta_1(y + \xi_p)}{\theta_2(0)} \left( \operatorname{Res} \Omega(z) \frac{d(z)}{\chi_\nu(z)} \Big|_{z=-y^*} + \operatorname{Res} \Omega(z) \frac{d(z)}{\chi_\nu(z)} \Big|_{z=\xi_p+1/2} \right). \tag{5.29}$$

Evaluating the residues in the rhs of (5.29) we find

$$\sum_{k=1}^{n} \mathcal{G}_k^z = -\mathcal{G}_{0;0}^z - \mathcal{G}_{0;+}^z. \tag{5.30}$$

Thus, we have proved that the form factor $\mathbf{F}_z^{\nu;\epsilon}(\bar{v}|\bar{v})$ vanishes independently of the value of $\epsilon$. Hence, $\mathbf{F}_z^{\nu,\nu}(\bar{v}|\bar{v}) = 0$, which implies $\mathcal{F}_{z,p}^{\nu,\nu}(\bar{v}|\bar{v}) = 0$.

## 5.2 Transversal form factor

We consider the form factor $\mathbf{F}_x^{\nu,\lambda}(\bar{v}|\bar{u})$ with $\nu = 0$ and $\lambda = 1$ for definiteness. As before, we introduce

$$\mathbf{F}_x^{0;\epsilon}(\bar{v}|\bar{u}) = \mathbf{F}_x^{0,1}(\bar{v}|\bar{u}) + (-1)^\epsilon \mathbf{F}_x^{0,3}(\bar{v}|\bar{u}), \qquad \epsilon = 0, 1. \tag{5.31}$$

Using the formulas of appendices B and C, we obtain explicit expressions for all three contributions in formula (3.9). We do not present the details of these calculations, since they are straightforward and completely analogous to the calculations of the previous section. We only note that we use identities for theta functions of the form (A.4), (A.14).

Ultimately, we present the form factor $\mathbf{F}^{\nu;\epsilon}(\bar{v}|\bar{u})$ as follows:

$$\mathbf{F}^{\nu;\epsilon}(\bar{v}|\bar{u}) = H_\epsilon G_\epsilon. \tag{5.32}$$

Here

$$H_\epsilon = i^\epsilon a(\xi) f(\bar{u}, \xi) S^0(\bar{v}|\bar{u}) \frac{\theta_4(0)\theta_2(0)}{2\theta_1(y + \xi)\theta_2(y + \xi)\theta_1(x)\theta_2(x)}, \tag{5.33}$$

where $S^0(\bar{v}|\bar{u})$ is given by (C.3). The function $G_\epsilon$ has the following form:

$$G_\epsilon = \theta_1(r + \xi_p + s_\epsilon)\theta_3(t_\epsilon - \xi_p) \frac{\theta_2(y^* + \bar{u})\theta_2(\xi_p - \bar{v})}{\theta_2(y^* + \bar{v})\theta_2(\xi_p - \bar{u})}$$

$$+ i \frac{\theta_1(\xi_p - \bar{u})\theta_1(y^* + \bar{v})}{\theta_1(\xi_p - \bar{v})\theta_1(y^* + \bar{u})} \theta_1(r - \xi_p + t_\epsilon)\theta_3(s_\epsilon + \xi_p) \frac{\chi_1(-y^*)}{\chi_0(-y^*)}. \tag{5.34}$$

where $r = \sum_{j=1}^{n}(v_j - u_j)$. Substituting here $\chi_1(-y^*)$ and $\chi_0(-y^*)$ in the form (2.18) we obtain

$$G_\epsilon = \frac{\theta_2(y^* + \bar{u})\theta_2(\xi_p - \bar{v})}{\theta_2(y^* + \bar{v})\theta_2(\xi_p - \bar{u})}\tilde{G}_\epsilon, \tag{5.35}$$

where

$$\tilde{G}_\epsilon = \theta_1(r + \xi_p + s_\epsilon)\theta_3(t_\epsilon - \xi_p) + e^{2\pi i(y^* + \xi_p)}\theta_1(r - \xi_p + t_\epsilon)\theta_3(s_\epsilon + \xi_p). \tag{5.36}$$

Using

$$\theta_3(z) = \theta_2(z - \tfrac{\tau}{2})e^{-i\pi(z - \tau/4)}, \tag{5.37}$$

and $2y^* = s - t - 1$ we transform (5.36) as follows:

$$\tilde{G}_\epsilon = e^{-i\pi(t_\epsilon - \xi_p - \tau/4)}\left(\theta_1(r + \xi_p + s_\epsilon)\theta_2(t_\epsilon - \xi_p - \tfrac{\tau}{2}) - \theta_1(r - \xi_p + t_\epsilon)\theta_2(s_\epsilon + \xi_p - \tfrac{\tau}{2})\right). \tag{5.38}$$

Observe that

$$e^{-i\pi_0 t_\epsilon} = e^{-i\pi(t + \epsilon/2)} = (-i)^\epsilon e^{i\pi t}. \tag{5.39}$$

Thus, we obtain the following representation for the form factor (5.32):

$$\mathbf{F}^{0;\epsilon}(\bar{v}|\bar{u}) = H_\epsilon^{\mathrm{mod}}G_\epsilon^{\mathrm{mod}}. \tag{5.40}$$

Here

$$H_\epsilon^{\mathrm{mod}} = a(\xi_p)f(\bar{u}, \xi_p)S^0(\bar{v}|\bar{u})\frac{e^{-i\pi(t - \xi_p - \tau/4)}\theta_4(0)\theta_2(0)}{2\theta_1(y + \xi_p)\theta_2(y + \xi_p)\theta_1(x)\theta_2(x)}\frac{\theta_1(y + \bar{u})\theta_2(\xi_p - \bar{v})}{\theta_1(y + \bar{v})\theta_2(\xi_p - \bar{u})}, \tag{5.41}$$

and

$$G_\epsilon^{\mathrm{mod}} = \theta_1(r + \xi_p + s_\epsilon)\theta_2(t_\epsilon - \xi_p - \tfrac{\tau}{2}) - \theta_1(r - \xi_p + t_\epsilon)\theta_2(s_\epsilon + \xi_p - \tfrac{\tau}{2}). \tag{5.42}$$

Equation (5.42) can be further simplified. Due to identity (A.6), we obtain

$$G_\epsilon^{\mathrm{mod}} = (-1)^\epsilon\theta_1(r + 2x - \tfrac{\tau}{2}|2\tau)\left(\theta_4(2\xi_p + 2y + r + \tfrac{\tau}{2}|2\tau) - \theta_4(2\xi_p + 2y - r - \tfrac{\tau}{2}|2\tau)\right)$$
$$+ \theta_4(r + 2x - \tfrac{\tau}{2}|2\tau)\left(\theta_1(2\xi_p + 2y + r + \tfrac{\tau}{2}|2\tau) + \theta_1(2\xi_p + 2y - r - \tfrac{\tau}{2}|2\tau)\right), \tag{5.43}$$

where we also used

$$\theta_1(2x_\epsilon + z|2\tau) = (-1)^\epsilon\theta_1(2x + z|2\tau), \qquad \theta_4(2x_\epsilon + z|2\tau) = \theta_4(2x + z|2\tau). \tag{5.44}$$

Due to the sum rule (2.14), we have

$$r = \tfrac{\mu_1}{2} - \tfrac{\tau}{2}, \tag{5.45}$$

where $\mu_1$ is an integer. Then

$$G_\epsilon^{\mathrm{mod}} = (-1)^\epsilon\theta_1(2x + \tfrac{\mu_1}{2} - \tau|2\tau)\left(\theta_4(2\xi_p + 2y + \tfrac{\mu_1}{2}|2\tau) - \theta_4(2\xi_p + 2y - \tfrac{\mu_1}{2}|2\tau)\right)$$
$$+ \theta_4(2x + \tfrac{\mu_1}{2} - \tau|2\tau)\left(\theta_1(2\xi_p + 2y + \tfrac{\mu_1}{2}|2\tau) + \theta_1(2\xi_p + 2y - \tfrac{\mu_1}{2}|2\tau)\right). \tag{5.46}$$

It remains to use

$$\theta_4(z + \tfrac{\mu_1}{2}|2\tau) - \theta_4(z - \tfrac{\mu_1}{2}|2\tau) = 0,$$
$$\theta_1(z + \tfrac{\mu_1}{2}|2\tau) + \theta_1(z - \tfrac{\mu_1}{2}|2\tau) = (i^{\mu_1} + (-i)^{\mu_1})\theta_1(z|2\tau). \tag{5.47}$$

This implies

$$G_\epsilon^{\mathrm{mod}} = -i\left(i^{\mu_1} + (-i)^{\mu_1}\right) e^{i\pi(2x-\tau/2)} \theta_1(2x|2\tau)\theta_1(2\xi_p + 2y|2\tau). \qquad (5.48)$$

Substituting this into (5.40) and using (A.7) we arrive at

$$\mathbf{F}^{0;\epsilon}(\bar{v}|\bar{u}) = -i\left(i^{\mu_1} + (-i)^{\mu_1}\right) a(\xi_p) f(\bar{u}, \xi_p) S^0(\bar{v}|\bar{u})$$
$$\times e^{i\pi(s+\xi_p-\tau/4)} \frac{\theta_4(0)\theta_2(0)}{2\theta_4^2(0|2\tau)} \frac{\theta_1(y+\bar{u})\theta_2(\xi_p-\bar{v})}{\theta_1(y+\bar{v})\theta_2(\xi_p-\bar{u})}. \qquad (5.49)$$

Since the result does not depend on $\epsilon$, we conclude that

$$\mathbf{F}^{0,1}(\bar{v}|\bar{u}) = \mathbf{F}^{0;\epsilon}(\bar{v}|\bar{u}), \qquad \mathbf{F}^{0,3}(\bar{v}|\bar{u}) = 0. \qquad (5.50)$$

Thus, we come to representation (4.4).

# Conclusion

In this paper, we have obtained explicit formulas for the form factors of local operators in the $XY$ model. We used for this the generalized algebraic Bethe ansatz since the $XY$ model possesses the 8-vertex $R$-matrix. However, the general calculation scheme remains the same as when using the standard algebraic Bethe ansatz. It includes the explicit solution of the quantum inverse problem, the calculation of the action of the monodromy matrix elements on the Bethe vectors, and the calculation of the resulting scalar products.

The last stage is the most technically difficult. That is why in this paper, we have limited ourselves to special cases of singlet states of the $XY$ chain. In other cases, explicit representations are not yet known for all scalar products needed to compute the form factors. However, this obstacle is purely technical. In [1], we described a method that allows one to obtain a system of linear equations for the scalar products of Bethe vectors in the case of an arbitrary rational value of the parameter $\eta$. Having solved this system, we will be able to calculate the form factors of local operators in the more general case of the $XYZ$ chain. We plan to address this issue in our future publications.

# Acknowledgements

We are grateful to A. Zabrodin and A. Zotov for numerous and fruitful discussions. The work of G.K. was supported by the SIMC postdoctoral grant of the Steklov Mathematical Institute. Section 5.2 of the paper was performed by N.S. The work of N.S. was supported by the Russian Science Foundation under grant no.19-11-00062, https://rscf.ru/en/project/19-11-00062/ , and performed at Steklov Mathematical Institute of Russian Academy of Sciences.

# A    Jacobi theta functions

Here we only give some basic properties of Jacobi theta functions used in the paper. See [35] for more details.

The Jacobi theta functions are defined as follows:

$$\theta_1(u|\tau) = -i\sum_{k\in\mathbb{Z}}(-1)^k q^{(k+\frac{1}{2})^2} e^{\pi i(2k+1)u},$$

$$\theta_2(u|\tau) = \sum_{k\in\mathbb{Z}} q^{(k+\frac{1}{2})^2} e^{\pi i(2k+1)u},$$

$$\theta_3(u|\tau) = \sum_{k\in\mathbb{Z}} q^{k^2} e^{2\pi iku},$$
(A.1)

$$\theta_4(u|\tau) = \sum_{k\in\mathbb{Z}}(-1)^k q^{k^2} e^{2\pi iku},$$

where $\tau \in \mathbb{C}$, $\Im\tau > 0$, and $q = e^{\pi i\tau}$.

Theta functions $\theta_a(u|\tau)$ with $a > 1$ can be obtained from $\theta_1(u|\tau)$ by shifts of the argument

$$\theta_2(u|\tau) = \theta_1(u + \tfrac{1}{2}|\tau),$$
$$\theta_3(u|\tau) = e^{i\pi(u+\tau/4)}\theta_1(u + \tfrac{1}{2} + \tfrac{\tau}{2}|\tau),$$
(A.2)
$$\theta_4(u|\tau) = -ie^{i\pi(u+\tau/4)}\theta_1(u + \tfrac{\tau}{2}|\tau).$$

The following shift properties are important:

$$\theta_1(u + 1|\tau) = -\theta_1(u|\tau), \qquad \theta_1(u + \tau|\tau) = -e^{-\pi i(2u+\tau)}\theta_1(u|\tau).$$
(A.3)

Properties of $\theta_a(u|\tau)$ with $a > 1$ with respect to the shifts by 1 and $\tau$ follow from (A.2).

Theta functions satisfy numerous identities based on periodicity [35]. We use some of them in this work. In particular, when calculating the form factor of $\sigma_p^x$, we use

$$\frac{\theta_1(u + v)\theta_3(u - v) - \theta_1(u - v)\theta_3(u + v)}{\theta_2(u)\theta_4(u)} = 2\frac{\theta_1(v)\theta_3(v)}{\theta_2(0)\theta_4(0)}.$$
(A.4)

To prove (A.4), it suffices to note that the lhs of this equation is a double periodic function of $u$ that has no poles in the fundamental domain. Therefore, this function is identically equal to a constant. Setting $u = 0$, we arrive at (A.4).

Similarly, one can prove an identity

$$2\theta_1(u + v|2\tau)\theta_4(u - v|2\tau) = \theta_1(u|\tau)\theta_2(v|\tau) + \theta_2(u|\tau)\theta_1(v|\tau).$$
(A.5)

It follows from this identity that

$$\theta_1(u|\tau)\theta_2(v|\tau) = \theta_1(u + v|2\tau)\theta_4(u - v|2\tau) + \theta_4(u + v|2\tau)\theta_1(u - v|2\tau).$$
(A.6)

In particular, setting $v = u$ in (A.6) we obtain

$$\theta_1(u|\tau)\theta_2(u|\tau) = \theta_1(2u|2\tau)\theta_4(0|2\tau).$$
(A.7)

It is often convenient to formulate the property of double periodicity as follows. Let $\Phi(z)$ be a double periodic function with periods 1 and $\tau$ and simple poles at $z = w_k$, $k = 1, \ldots, r$, in the fundamental domain. Then

$$\sum_{k=1}^{r} \mathrm{Res}\,\Phi(z)\Big|_{z=w_k} = 0.$$
(A.8)

This identity immediately follows from the fact that

$$\oint \Phi(z)\,\mathrm{d}z = 0, \tag{A.9}$$

where the integral is taken along the boundary of the fundamental domain.

In particular, when calculating the form factor of $\sigma_p^z$, we use the following identities:

$$\theta_1(x_\epsilon)\theta_2(y + \xi_p)\theta_2(v_k + y)\theta_1(v_k - \xi_p - x_\epsilon) - \theta_2(0)\theta_1(s_\epsilon + \xi_p)\theta_1(v_k - t_\epsilon)\theta_2(v_k - \xi_p)$$
$$= -\theta_2(x_\epsilon)\theta_1(y + \xi_p)\theta_1(v_k + y)\theta_2(v_k - \xi_p - x_\epsilon), \quad \text{(A.10)}$$

and

$$\theta_1(x_\epsilon)\theta_2(y + \xi_p)\theta_1(v_k + y)\theta_2(v_k - \xi_p + x_\epsilon) - \theta_2(0)\theta_1(t_\epsilon - \xi_p)\theta_2(v_k + s_\epsilon)\theta_1(v_k - \xi_p)$$
$$= \theta_2(x_\epsilon)\theta_1(y + \xi_p)\theta_2(v_k + y)\theta_1(v_k - \xi_p + x_\epsilon). \quad \text{(A.11)}$$

These identities respectively follow from

$$\oint \frac{\theta_1(z + x_\epsilon)\theta_2(z + y + \xi_p)\theta_1(z + v_k - \xi_p - x_\epsilon)}{\theta_1(z)\theta_2(z)\theta_1(z + v_k + y)}\,\mathrm{d}z = 0, \tag{A.12}$$

and

$$\oint \frac{\theta_1(z - x_\epsilon)\theta_2(z + y + \xi_p)\theta_2(z + v_k - \xi_p + x_\epsilon)}{\theta_1(z)\theta_2(z)\theta_2(z + v_k + y)}\,\mathrm{d}z = 0. \tag{A.13}$$

Let us also give an example of a more sophisticated identity, which is used in the calculation of transversal form factors:

$$\sum_{k=1}^{n} \frac{\theta_1(u_k - \xi_p - x_\epsilon)\theta_2(r + u_k + s_\epsilon)}{\theta_2(u_k - \xi_p)\theta_1(u_k + y)} \frac{\theta_1(u_k - \bar{v})}{\theta_1(u_k - \bar{u}_k)}$$
$$= \frac{\theta_1(\xi_p + s_\epsilon)\theta_2(r + x_\epsilon)}{\theta_2(y + \xi_p)} \frac{\theta_1(y + \bar{v})}{\theta_1(y + \bar{u})} - \frac{\theta_2(x_\epsilon)\theta_1(r + \xi_p + s_\epsilon)}{\theta_2(\xi_p + y)} \frac{\theta_2(\xi_p - \bar{v})}{\theta_2(\xi_p - \bar{u})}. \quad \text{(A.14)}$$

Here $r = \sum_{j=1}^{n}(v_j - u_j)$. This identity follows from the analysis of a contour integral

$$\oint \frac{\theta_1(z - \xi_p - x_\epsilon)\theta_2(r + z + s_\epsilon)}{\theta_2(z - \xi_p)\theta_1(z + y)} \frac{\theta_1(z - \bar{v})}{\theta_1(z - \bar{u})}\,\mathrm{d}z = 0. \tag{A.15}$$

The sum of the residues in $z = u_k$, $k = 1,\ldots,n$, gives the lhs of (A.14). The residues in $z = \xi_p + 1/2$ and $z = -y$ give the rhs of (A.14).

.

# B   Coefficients of the action formula

To describe the coefficients $\mathbf{W}_{a;0}^{(\lambda,\mu)}(w_{n+1}, w_k)$ we introduce three functions

$$\begin{aligned}
\alpha_l^x(z) &= (-1)^l \theta_4(y + w_{n+1})\theta_3(x_l)\theta_1(z + x_l), \\
\alpha_l^y(z) &= i(-1)^l \theta_3(y + w_{n+1})\theta_4(x_l)\theta_1(z + x_l), \\
\alpha_l^z(z) &= (-1)^l \theta_2(y + w_{n+1})\theta_1(x_l)\theta_1(z + x_l),
\end{aligned} \tag{B.1}$$

and their Fourier transforms

$$\hat{\alpha}_\mu^a(z) = \sum_{l=0}^3 e^{-i\pi\mu l/2}\alpha_l^a(z), \qquad a \in \{x,y,z\}. \tag{B.2}$$

Then

$$\mathbf{W}_{a;0}^{(\lambda,\mu)}(w_{n+1},w_k) = \frac{\hat{\alpha}_{\lambda-\mu}^a(w_{n+1}-w_k)f(w_k,\bar{w}_k)\Big((-1)^n e^{i\pi\mu/2}a(w_k) - e^{-i\pi\mu/2}d(w_k)\Big)}{4\theta_1(y+w_{n+1})h(w_k,w_{n+1})\theta_1(x)\theta_2(x)}. \tag{B.3}$$

To describe the coefficients $\mathbf{W}_{a;-}^{(\lambda-\mu)}(w_{n+1},w_k,w_l)$ we introduce three functions

$$\begin{aligned}
\beta_l^x(z,w_j,w_k) &= \theta_4(0)\theta_3(t_l-z)\theta_1(z-w_j+x_l)\theta_1(z-w_k+x_l),\\
\beta_l^y(z,w_j,w_k) &= i\theta_3(0)\theta_4(t_l-z)\theta_1(z-w_j+x_l)\theta_1(z-w_k+x_l),\\
\beta_l^z(z,w_j,w_k) &= \theta_2(0)\theta_1(t_l-z)\theta_1(z-w_j+x_l)\theta_1(z-w_k+x_l),
\end{aligned} \tag{B.4}$$

and their Fourier transforms

$$\hat{\beta}_\mu^a(z,w_j,w_k) = \sum_{l=0}^3 e^{-i\pi\mu l/2}\beta_l^a(z,w_j,w_k), \qquad a \in \{x,y,z\}. \tag{B.5}$$

Then

$$\mathbf{W}_{a;-}^{(\lambda-\mu)}(w_{n+1},w_k,w_l) = \frac{\hat{\beta}_{\lambda-\mu}^a(w_{n+1},w_k,w_l)\omega_{kl}(w_{n+1})}{2\theta_1(y+w_{n+1})\theta_2(0)\theta_1^2(x)\theta_2^2(x)}, \tag{B.6}$$

where

$$\omega_{kl}(z) = \frac{[d(w_k)a(w_l) - d(w_l)a(w_k)]\,f(w_k,\bar{w}_k)f(\bar{w}_l,w_l)}{f(w_k,w_l)h(w_k,z)h(z,w_l)}. \tag{B.7}$$

Finally, to describe the coefficients $\mathbf{W}_{a;-}^{(\lambda-\mu)}(w_{n+1})$ we introduce three functions

$$\begin{aligned}
\gamma_l^x(z) &= \theta_4(0)\theta_3(s_l+z),\\
\gamma_l^y(z) &= i\theta_3(0)\theta_4(s_l+z),\\
\gamma_l^z(z) &= \theta_2(0)\theta_1(s_l+z),
\end{aligned} \tag{B.8}$$

and their Fourier transforms

$$\hat{\gamma}_\mu^a(z) = \sum_{l=0}^3 e^{-i\pi\mu l/2}\gamma_l^a(z), \qquad a \in \{x,y,z\}. \tag{B.9}$$

Then

$$\mathbf{W}_{a;+}^{(\lambda-\mu)}(w_{n+1}) = -\frac{\theta_2(0)\hat{\gamma}_{\lambda-\mu}^a(w_{n+1})}{8\theta_1(y+w_{n+1})}. \tag{B.10}$$

# C  Scalar products

Scalar products (3.3) satisfy a selection rule

$$\mathbf{S}_{n,m}^{\nu,\lambda}(\bar{v}|\bar{w}) \cong \delta_{\nu+n,\lambda+m \,(\mathrm{mod}\,2)}\,. \tag{C.1}$$

In the case of free fermions, it follows from (C.1) that either $\lambda = \nu$ or $\lambda = \nu + 2 \bmod 2$.

It is convenient to introduce

$$\mathbf{S}_{n,m}^{\nu;\epsilon}(\bar{v}|\bar{w}) = \mathbf{S}_{n,m}^{\nu,\lambda}(\bar{v}|\bar{w}) + (-1)^{\epsilon}\mathbf{S}_{n,m}^{\nu,\lambda+2}(\bar{v}|\bar{w}), \qquad \epsilon = 0, 1. \tag{C.2}$$

Due to the selection rule, non-vanishing scalar products $\mathbf{S}_{n,m}^{\nu;\epsilon}(\bar{v}|\bar{w})$ occur for either $\lambda = \nu$ for $m = n$ or $\lambda = \nu + 1$ for $m = n \pm 1$.

We first give an explicit expression for $S_{n,n}^{\nu;\epsilon}(\bar{v}|\bar{w})$ (see [16]). Let

$$S^{\nu}(\bar{v}|\bar{w}) = \frac{\left(-e^{-i\pi\nu/2}\theta_1'(0)\right)^n}{\prod_{\substack{a,b=1 \\ a\neq b}}^{n} f(v_a, v_b)} \left(\prod_{k=1}^{n} \frac{\chi_{\nu}(w_k)}{a(v_k)\mathcal{V}_k}\right) \frac{\prod_{a>b}^{n} \theta_2(v_a - v_b)\theta_2(w_a - w_b)}{\prod_{a=1}^{n} \prod_{b=1}^{n} \theta_1(w_a - v_b)}. \tag{C.3}$$

where $\mathcal{V}_k$ is given by (4.2). Then

$$S_{n,n}^{\nu;\epsilon}(\bar{v}|\bar{w}) = \frac{\theta_1(r + x_\epsilon)}{\theta_1(x_\epsilon)} S^{\nu}(\bar{v}|\bar{w}), \tag{C.4}$$

where

$$r = \sum_{j=1}^{n}(v_j - w_j). \tag{C.5}$$

Scalar products $S_{n,n\pm1}^{\nu;\epsilon}$ can be expressed in terms of $S_{n,n}^{\nu;\epsilon}$. Let $\bar{w} = \{w_1, \ldots, w_{n-1}\}$. Then

$$\mathbf{S}_{n,n-1}^{\nu;\epsilon}(\bar{v}|\bar{w}) = -(-i)^{\epsilon} \frac{\theta_2(0)\theta_1(x_\epsilon)}{2T_{\nu}(-y^*|\bar{v})} \mathbf{S}_{n,n}^{\nu;\epsilon}(\bar{v}|\{\bar{w}, -y^*\}), \tag{C.6}$$

where $y^* = y - 1/2$.

Let now $\bar{w} = \{w_1, \ldots, w_{n+1}\}$. Then

$$\mathbf{S}_{n,n+1}^{\nu;\epsilon}(\bar{v}|\bar{w}) = \sum_{a>b}^{n+2} \frac{-2(-i)^{\epsilon}\theta_1(x_\epsilon)\omega_{ab}(-y^*)}{\theta_1^2(x)\theta_2^2(x)\theta_2(0)T_{\nu}(-y^*|\bar{v})} \theta_1(w_a' - t_\epsilon)\theta_1(w_b' - t_\epsilon)\mathbf{S}_{n,n}^{\nu;\epsilon}(\bar{v}|\bar{w}_{a,b}'), \tag{C.7}$$

where $\bar{w}' = \{\bar{w}, -y^*\}$ and the coefficients $\omega_{ab}$ are given by (B.7).

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
