# Peer review of "Form factor of local operators in the generalized algebraic Bethe ansatz"

_SciPost Physics_

## Round 2 · Referee Report · Anonymous · 2023-10-4

Report

This paper is a continuation of research performed by the authors in the papers [1,15] and by the second author in [16] according to the reference list of the paper. It is devoted to the investigation of $XYZ$ spin chain in the framework of the generalized algebraic Bethe ansatz. Form factors of local operators are studied in the limit of of free fermions, when this model becomes $XY$ spin chain.

The paper is rather technical and will require certain work from the reader to perform all missed calculations. Although the authors do not set themselves the goal to obtain new results, the importance of this paper is a demonstration of the fundamental
applicability of the generalized Bethe ansatz to the calculation of the form factors. In this respect, the paper meets expectations and criteria for the SciPost Physics Journal and I would like to recommend the paper ”Form factor of local operators
in the generalized algebraic Bethe ansatz” by G. Kulkarni and N.A. Slavnov for publication in SciPost Physics Journal.

Requested changes

The only remark which I would like to mention is a possible misunderstanding of the reader at the beginning of the section 2.2.1. The authors explicitly mentioned after formulas (2.7) that they are focusing in the paper on the case of $XY$ model, but
I think that it is reasonable to mention this again somewhere around the Bethe equations (2.13), which take in this case very specific and decoupled form.

Attachment

---

## Round 2 · Referee Report · Jules Lamers · 2023-10-25

Strengths

- The authors compute form factors for certain (singlet) states for a chain of even length at the free-fermion point (XY model).
- The resulting expressions are explicit.
- Even if the paper does not set out to derive new results, it presents progress towards the important problem of calculating correlation functions for the Heisenberg XYZ chain.

Weaknesses

- The paper would benefit from a more detailed description of the context and scope of the results.
- The paper is not so easy to understand without going back to previous papers of the authors.
- The authors can clarify which results were previously known, with referece, and which are new.
- The work of Felder et al seems to be essentially ignored (here and in prior works of the authors on this topic).

Report

The authors continue their form-factor approach to correlation functions of local operators for the Heisenberg XYZ chain from Refs [16, 15, 1] using the 'generalised algebraic Bethe ansatz'. Specifically, in the present paper, the authors compute form factors (i.e. one-point correlation functions with \sigma^x,y,z) for singlet states for a chain of even length at the free-fermion point (XY model), obtaining explicit results.

The calculation of correlation functions for the Heisenberg XYZ chain is an important and long-standing problem. With this in mind, the results in the paper for the (rather) simpler XY model are interesting too, especially as a step towards and test/check for the general case. At the same time, the paper is not so easy to understand without going back to previous work of the authors, and would benefit from a more detailed description of the context and scope of the results. I recommend revisions before publication.

Requested changes

I suggest the authors improve the exposition. They should clarify the scope of their results, preferably qualitatively in the introduction and in more detail in section 2. Specifically, please
1. Explain more clearly what singlet eigenstates are and what twin-free eigenstates are. How many such states are there (at least qualitatively)? Do they includes the groundstate? And low-lying excitations? In which phase? (On p6 it seems to be suggested that twin-free states are the same as singlet states; is that so?)
2. Explain more clearly which correlations can be obtained using form factors with the non-standard normalisation (3.2), and especially which ones (if any) can not be obtained in this way.

In addition, the authors should more carefully compare their works with the literature.
3. In sections 4 and 5 it would be useful to clarify which results were earlier obtained (with references) and which are new.

More importantly, the work of Felder and Varchenko (1996, see arXiv:q-alg/9601003 and especially q-alg/9605024) should be mentioned. This is especially true for the material of references [16], [15] and [1], which should already have cited and compared their results to that of Felder and Varchenko.
(Indeed, reference [15] appears to have picked up directly after the work of Takhtajan and Faddeev from 1979 while disregarding or unaware of decades of relevant subsequent developments by Felder et al. In [15] the terms 'dynamical R-matrices' and '8vSOS model' appear once, and Felder is cited once. Moreover, none of this is mentioned in the earlier paper [16], in [1] or the present paper. Yet Felder's elliptic quantum group is the proper algebraic setting for those works. In particular, the 'generalised gauge-transformed monodromy matrix' is just the dynamical monodromy matrix, and the 'generalised algebraic Bethe ansatz' is just the algebraic Bethe ansatz using the dynamical B-operator. Incidentally, I believe that the more modern notation of Felder et al also makes the subscripts _{l,l+N} etc of the dynamical A-, ..., D-operators in [16,15,1] unnecessary.)

4. In the present paper, the authors should at the very least in their conclusion comment on the relation of their work to that of Felder and Varchenko. In addition, I hope that they will read up and properly mention and cite those works in their future work.

Other aspects of the exposition that can be improved are:
5. In section 2.2.1, explain the meaning/origin of the parameter \nu (and why it is an integer mod 4 for free fermions), of the sum rule (2.14), and of the parameters s and t from (2.24). (Are the latter related to the dynamical parameter, or something else?)
6. Grammar - see some examples below.

Below follow various minor comments and suggestions.

Page 1
- Perhaps the title should be "Form factor_s_ ..." (plural); I am not sure.
- Incorrect articles: preceding (1.1) "_The_ Hamiltonian", following (1.1) "_the_ Hilbert space", a little further "_the_ 8-vertex model".

Page 3
- Incorrect articles: preceding (2.1) "_the_ Hamiltonian", preceding (2.5) "_the_ homogeneous case", following (2.6) "_the_ transfer matrix", following (2.7) "_the_ XY chain", next line "_the_ homogeneous case".
- Superfluous articles preceding (2.4): "the XYZ chain of _ length N is equal to a product of _ R-matrices"
- Missing word following (2.1): "the complex _parameter/variable_ u"

Please try to check the article usage throughout.

Page 4
- Typo preceding (2.11): "parameterized _by_ a set of complex numbers"

Page 5
- Top: clarify what is meant by "where \nu_1 takes integer values" - should there exist an integer \nu_1 such that (2.14) holds, or should (2.14) hold for any integer \nu_1?
- What will give the usual rapidities in the XXZ limit - u (as one might naively expect), i u (since the Fabricius-McCoy strings seem to be spaced by a 1/2, i.e. in the real rather than imaginary direction), ...?
- Preceding (2.17) there seems to be some confusion: whilst closely related to the torus \mathbb{C}/(\matbb{Z} + \tau \mathbb{Z}), the fundamental domain is a subset of \mathbb{C} representing that quotient. For instance, the torus has no boundary, whereas following (5.28) the authors clearly want the fundamental domain to have nonempty boundary.
- Following (2.17), should the condition be 0 \leq \Re(z mod 1) < 1/2 ?
- Missing words following (2.17): "is a root _of_ \chi_1(z)" (twice)
- Preceding Prop 2.1 I think "any" should be "all"
- In the proof, "the equation (2.11)" cannot be a polynomial, but "the function (2.11)" can be.
- Perhaps explain or give a reference for the notion of 'elliptic polynomial' as this is less well known or obvious than in the rational and trigonometric cases.
- In the next line, does "First" mean "Among them"?

Page 6
- To explain what s and t mean, it is probably better to include the face/vertex transformation. I am familiar with the latter, but not with your notation; the current sentence does not explain to me where s,t come from or why the dependence on them can be factored out.
- Following (2.25), does "the singlet eigenstate" mean "_any_ singlet eigenstate" or only some of them? Please clarify.
- What does "[The dual Bethe vectors] are arranged in a completely similar way to the Bethe vectors described above" mean? (The vectors were only described, but not given. Everything starts to become rather vague.)
- Section 3: "the singlet part of the spectrum, that is, twin-free states" - are these two notions the same? That was not at all clear from the relevant parts of Section 2.
- The reason given preceding (3.3) is not a real reason: explain why this non-standard ('special') normalisation was used in [1, 16].
- Around (3.4) the discussion is somewhat vague: "expressions that are quadratic in form factors, for instance ... such expressions turn out..." suggests that any expression quadratic in form factors would be normalised in the standard way, but is this also true if the two form factors have different parameters \nu,\lambda,\bar{u},\bar{v}? The subsequent phrasing "they should not" does not make it quite clear whether or when the result really is independent of the gauge transformation.

Page 7
- Concerning footnote 3, since (3.5) is obvious graphically provided T(u)^{-1} is equal to the opposite transfer matrix, I was wondering if the latter is true and, if so, easy to show?
- Perhaps clarify that \bar{w} is (I think) equal to \bar{u} with \xi_p appended (and this is why in [15] you first needed to extend [1] to unbalanced scalar products).

Page 8
- It is not directly clear how sections 4 and 5 are related - section 5.2 seems to contain a sketch of the computation leading to (4.4) in section 4.2, but section 5.1 seems to establish a result whose link to section 4.1 is less clear at a glance.

While I did not check the computations in Section 5 in detail, the results look reasonable.

Page 12
- Does "we present the form factor as follows" mean "we express/write/... the form factor as follows"?

Page 17
- Preceding (B.8) the subscript of W should be a + ?
- What is meant by the \simeq in (C.1); proportionality? Should \nu+n in the first argument of the Kronecker symbol also be mod 2?

Page 18
- There are four instances of S that should perhaps be \mathbf{S}?

---

## Editorial Decision

awaiting_resubmission